# Super-Resolution and Wide-Field-of-View Imaging Based on Large-Angle Deflection with Risley Prisms

**DOI:** 10.3390/s23041793

**Published:** 2023-02-05

**Authors:** Chao Gui, Detian Wang, Xiwang Huang, Chunyan Wu, Xin Chen, Huachuan Huang

**Affiliations:** 1Key Laboratory of Testing Technology for Manufacturing Process, School of Manufacturing Science and Engineering, Southwest University of Science and Technology, Mianyang 621010, China; 2Institute of Fluid Physics, China Academy of Engineering Physics, Mianyang 621900, China; 3School of Computer Science and Technology, Southwest University of Science and Technology, Mianyang 621010, China

**Keywords:** distortion correction, super-resolution reconstruction, field of view extension

## Abstract

A novel single camera combined with Risley prisms is proposed to achieve a super-resolution (SR) imaging and field-of-view extension (FOV) imaging method. We develop a mathematical model to consider the imaging aberrations caused by large-angle beam deflection and propose an SR reconstruction scheme that uses a beam backtracking method for image correction combined with a sub-pixel shift alignment technique. For the FOV extension, we provide a new scheme for the scanning position path of the Risley prisms and the number of image acquisitions, which improves the acquisition efficiency and reduces the complexity of image stitching. Simulation results show that the method can increase the image resolution to the diffraction limit of the optical system for imaging systems where the resolution is limited by the pixel size. Experimental results and analytical verification yield that the resolution of the image can be improved by a factor of 2.5, and the FOV extended by a factor of 3 at a reconstruction factor of 5. The FOV extension is in general agreement with the simulation results. Risley prisms can provide a more general, low-cost, and efficient method for SR reconstruction, FOV expansion, central concave imaging, and various scanning imaging.

## 1. Introduction

The imaging FOV and resolution are key parameters for performance evaluation in various machine vision applications, which play an important role in remote sensing, infrared reconnaissance, and medical-related fields (e.g., tissue sectioning, and dynamic observation of living cells) [1,2]. For most conventional vision systems, achieving large FOV and high-resolution (HR) is challenging because these two requirements are opposed to each other. The limitation of image resolution comes from two aspects. On the one hand, it is a limitation on the sensor resolution enhancement because the pixel size of image sensors cannot be infinitely small as far as manufacturing processing is concerned, and the resolution of many commercial cameras is limited by the pixel mix, which is much more expensive and difficult to manufacture. On the other hand, the image resolution is limited by the limitation of optical diffraction limit, which cannot make the imaging image point infinitely small. Techniques to achieve optical diffraction limit breakthrough can be divided into two main categories; one is based on single-molecule localization imaging methods, which use the optical switching properties of special fluorescent molecules or other mechanisms to randomly excite sparse fluorescent dots at different moments and localize the fluorescent molecules by corresponding algorithms [3,4,5], and the other is to change the point spread function in the imaging system, which in turn enables imaging that breaks the diffraction limit resolution [6]. SR reconstruction techniques have been shown to be an effective method for improving the spatial resolution to the diffraction limit of an optical system [7,8]. In this paper, we focus on the use of Risley prisms to acquire subpixel images to achieve sensor resolution enhancement.SR reconstruction technique is the process of recovering low-resolution (LR) images to HR images by digital image processing algorithms, which can be classified into two types, single-frame images and multi-frame images, depending on the number of original LR images The single-frame SR reconstruction technique is to establish the a priori relationship between LR and HR images in advance, i.e., to establish a mathematical model to achieve SR reconstruction, which is cost-effective and lacks generality. The SR reconstruction technique of multi-frame images is to recover the sub-pixel displacement of multiple LR images relative to the scene motion in the same scene by reconstruction algorithm to obtain HR images by extracting and integrating more details from multiple LR images, so it does not require a priori knowledge and is more widespread and generalized compared to the SR reconstruction technique of single images [9,10,11].

For SR reconstruction techniques of multi-frame images, the sub-pixel imaging technique becomes one of the key techniques to obtain HR images. The current common methods to obtain sub-pixel shifted image sequences are the sub-imaging method and the microscan method. The meaning of the sub-imaging method is to obtain sub-pixel-shifted LR sequence images by changing the detector image element perception chip. For example, photoreceptor surface array imaging, a bionic-like array detector, this method has real-time characteristics to obtain sequential LR images in a single exposure to reconstruct HR images, however, it cannot expand the FOV of a single camera [7,12]. The microscan method achieves sub-pixel image acquisition by moving the detector or optical axis, which is performed by rotating or vibrating the optical elements of the imaging lens. There are other imaging methods, such as compact lens-free microscope imaging techniques, and digital inline holograms with sub-pixel shifts obtained by adjusting the angle through volumetric phase grating diffraction [13,14,15]. Carles [16] et al. used optical wedge prisms and camera arrays to achieve SR of the central field while achieving twice the FOV extension. It is more miniaturized, compact, interference-resistant, and economical compared to the array camera, a single camera combined with the Risley prisms imaging system (RPIS), which enables FOV expansion of a single camera compared to sub-imaging and lens-free holography.

Scanning imaging for FOV extension is based on a rotating stage and a reflector assembly. Mounting a camera on a rotating stage creates a large FOV image [17], and placing a camera in front of a reflector assembly enables FOV extension by continuous light deviation [18]. The following scanners with various mirror assemblies, such as flow detector-based scanners [19], multifaceted mirror scanners [20], and microelectromechanical systems [21]. They have been widely used in biomedical imaging. Unfortunately, these mechanical or optical reflection scanners are limited in many application scenarios due to their large physical size, relative sensitivity to mechanical errors, and inherent rotational inertia. In contrast, Risley prisms scanners, which are small, compact, and have superior dynamic performance, can be consulted for studies on FOV expansion, including a comprehensive analysis of scanning patterns and accurately modeled graphical solutions. In fact, Risley prisms have been used for precise pointing of the camera imaging view axis to perform stepped gaze imaging functions. Recent work has also reported the combination of Risley prismatic scanners with artificial compound eyes to achieve SR imaging in a certain region.

Risley prisms are used to change the imaging visual axis because of their precise pointing, fast response frequency, high interference immunity, and low power consumption [22]. Therefore, RPIS can be used for FOV extension and sub-pixel image sequence acquisition. In this paper, we use RPIS to implement SR reconstruction and FOV extension. Firstly, an SR reconstruction system model consisting of a single camera and Risley prisms with a precisely adjusted visual axis is established. The SR reconstruction factor is determined based on the optical-related knowledge, and the FOV extension sub-region location and scan path are determined by the Risley prisms covering the FOV and the camera FOV. In this paper, the super-resolution imaging problem caused by large-angle deflection is solved by using a large-angle deflection Risley prisms combined with a single camera for super-resolution imaging, and the experimental results show that the super-resolution reconstruction effect in the case of large-angle beam deflection is comparable to that of the small-angle beam deflection of the Risley prisms in the literature [23,24]. In addition, this paper also analyzes the FOV expansion capability based on the imaging system parameters (e.g., Risley prisms-related parameters, camera, and lens parameters) and plans the expansion path, number, and order, and the FOV expansion results match well with the simulation results, which makes the Risley prisms combined with a single camera imaging system more popular and widespread. In conclusion, the method has a broad development prospect with the advantages of low cost, high efficiency, and high stability in realizing SR reconstruction, FOV expansion, and central concave imaging.

The rest of this paper is summarized as follows. In Section 2, the imaging distortion phenomenon and distortion correction principle due to the nonlinear characteristics of Risley prisms on beam deflection are analyzed, and the SR reconstruction principle and method and how to plan the FOV extension sub-region location are described. In Section 3, the feasibility of distortion correction and FOV extension is verified by simulation experimental results. In Section 4, conclusions are drawn by comparing experimental and simulation verification.

## 2. Methods

The RPIS system is shown in Figure 1a, and the prism structure with the combination 21-12 shown in Figure 1b is used in this paper, and the red solid line represents the imaging deflection optical axis of the camera, which is mainly represented by two parameters, namely the deflection angle E (the angle between the main optical axis z and the imaging deflection optical axis), and the azimuth angle A (the angle between the projection of the deflection optical axis in the target plane and the positive direction of the x-axis). The outgoing beam can be described in this way *A_r_4__* = [*K_r_4__*, *L_r_4__*, *M_r_4__*]*^T^* [25].
(1)$$E=\arccos(-M_{r_{4}})$$
(2)$$A=\left\{ \begin{array}{lll} \mathrm{undefined}\;\; & , & if\;K_{r_{4}}=0\;and\;L_{r_{4}}=0\\ \arctan(\frac{L_{r_{4}}}{K_{r_{4}}}) & , & \begin{array}{c} if\;K_{r_{4}}\geq 0\;and\;L_{r_{4}}>0\\ or\;K_{r_{4}}>0\;and\;L_{r_{4}}\geq 0 \end{array}\\ \arctan(\frac{L_{r_{4}}}{K_{r_{4}}})+2\pi \;\; & , & if\;K_{r_{4}}\geq 0\;and\;L_{r_{4}}<0\\ \arctan(\frac{L}{K_{r_{4}}})+\pi \;\; & , & otherwise \end{array}\right.$$

### 2.1. Image Distortion and Correction

Ideal pinhole imaging is achieved when the imaging axis is parallel to the main optical axis of the Risley prisms, and the incident light vector of the object is projected onto the image plane through the pinhole camera model without distortion, and the incident light vector is obtained from Equation (3).
(3)$$I_{o}=\frac{{\left[x_{k},y_{k},z_{k}\right]}^{T}}{\sqrt{x_{k}^{2}+y_{k}^{2}+z_{k}^{2}}}$$
where $\left(x_{k},y_{k},z_{k}\right)$ denotes the position of the pixel point under the camera coordinate system, $x_{k}=\left(V-O_{x}\right)\cdot p$, $y_{k}=\left(U-O_{x}\right)\cdot p$, $z_{k}=f$, *p* is the pixel pitch, $\left(O_{x},O_{y}\right)$ is the center of the optical axis under the camera coordinates, and (U,V) denotes that a pixel point is located in a matrix of M×N pixel size.

The imaging distortions caused by the nonlinear nature of the Risley prisms on the beam pointing when the imaging view axis is pointing to other positions result in a nonlinear change in the imaging FOV and resolution. To better demonstrate the image distortion phenomenon and distortion correction simulation results, we define two unit vectors, $L_{i}$ and $L_{r}$, as shown in Figure 2a. The vector $L_{i}$ is the arbitrary incident beam in the object position through the Risley prisms, and the vector $L_{r}$ is the corresponding outgoing beam, i.e., the receiving beam of the camera. Describing the relative rotation relation of the two spatial vectors can be described by the Euler-Rodrigues rotation equation M [26].

From the imaging principle, it can be derived that the received beam of the camera needs to be brought together to a point namely the focal point, before projecting to the image plane, mapped to a point on the image plane as z=f in Figure 2a, where *f* is denoted as the equivalent focal length, which is related to the beam deflection angle to the distorted incident beam of the camera can be described as:(4)$$L_{r}=L_{i}\cdot M$$

The nonlinear deflection of the object-side incident beam through the Risley prisms leads to compression and distortion of the image. Figure 2b shows the effect of different combinations of prism rotation angles on the imaging FOV, and it can be concluded that the imaging distortion is most severe when the imaging view axis is at the maximum deflection angle, and the distortion state is compressed along the view axis direction. When the viewpoint is at the central optical axis, the image is distortion-free. The principle of aberration correction is based on the reversibility of light, the original camera received beam $L_{r}$ into the emitted beam can restore the true physical size of the target object, effectively correcting the problem of imaging aberrations. [27].

### 2.2. Sub-Pixel Shift for Super-Resolution Imaging

Figure 3 depicts the principle of sub-pixel LR sequence images for SR reconstruction. Sub-pixel shift images are obtained using sub-pixel scanning imaging methods under RPIS. The principle of sub-pixel imaging is that the data of the imaged object is continuous, but the image data acquired by camera imaging is discretized due to the size limitation of individual pixels, and these pixels that exist between two physical pixels are called sub-pixels. Figure 3 illustrates that LR sequence image acquisition is performed with image 1 as the reference and all other images are sub-pixel shifted with it as the center.

The pixel resolution of current commercial cameras is greatly affected by factors such as sensor chip size, processing, and price. SR reconstruction techniques can increase the spatial resolution to the optical diffraction limit, and the Nyquist frequency of the image sensor is expressed as $v_{p}=1/2p$, where *p* is the pixel pitch. The optical spatial cutoff frequency in the diffraction limit is denoted as $v_{0}=D/1.22\lambda f$, where *D* is the diameter of the pupil of entry, *λ* is the optical wavelength, and *f* is the camera focal length. We assume that $H=v_{0}/v_{p}$, implying that the spatial resolution of the camera can be increased by a factor of *H* using SR reconstruction techniques. *H* can be considered as the degree of blending of the imaging system, which is a constant once the camera parameters (containing focal length *f*, aperture number ratio *F*, and pixel pitch *p*) as well as the imaging wavelength are determined. However, the reconstruction factor *h* < *H* in practical applications is due to processing, mounting, and calibration errors in the imaging system. When the reconstruction factor is *h*, the step of sub-pixel displacement in the image sensor is sl=p/h, and the step of sub-pixel displacement in the corresponding object plane is SL=sl×u/f. In RPIS, the intersection of the main optical axis and the image plane is fixed, and the imaging FOV varies with the imaging view axis rotation. Therefore, in order to acquire sub-pixel images, we only need to shift the imaging FOV in sub-pixel steps. The method of sub-pixel shift of the imaging FOV through the Risley prisms is described below.

First, a reconstruction factor is determined to calculate the rotation angles of the two prisms corresponding to the sub-pixel displacement acquisition positions. Figure 4 illustrates the pixel point cloud map of the imaging view axis in the object plane for the desired SR image with *h* = 5. The sequence of numbers represents the LR acquisition sequence. The key to sub-pixel imaging is the precise location of the imaging view axis points to avoid excessive image data redundancy. For the problem of how the camera acquires accurate sub-pixel images, we propose a solution to obtain the sub-pixel positions about the imaging view axis using the exact inverse solution and Newton’s iteration method. The following are the detailed steps for solving the sub-pixel acquisition position.

(a) First, determine the imaging position of the reference image (sequence 1) and the reconstruction factor h.

(b) In the second step, establish the object plane theoretical imaging view axis point cloud based on the step SL, the object distance *u*.

(c) In the third step, the imaging position of the reference image is used as the origin to calculate the azimuth angle A and deflection angle E at other theoretical positions of the object plane, and the exact inverse solution proposed by Alajlouni [28] combined with the Newtonian iterative method is used to calculate the prism’s turning angles θ_1_, θ_2_.

Then, in Figure 4, the sub-pixel scanning imaging is performed by the clockwise spiral scanning method with the scanning paths from point 1 to point 25 in sequence. Once the imaging view axis is scanned according to the desired path, the sequential image acquisition of the sub-pixel displacement is completed.

Finally, when the control biprism reaches a specified position, an LR image is obtained accordingly, and eventually h^2^ serial LR images are obtained. If these LR images were acquired by ideal acquisition in an ideal environment, the relative displacement between them would be known. However, due to errors in processing, assembly, and building the RPIS, image alignment must be performed on the sequence LR images before SR reconstruction is performed. Accelerated robust feature (SURF) is widely used in feature point detection, as well as in feature matching, etc. This algorithm can achieve sub-pixel image alignment, and compared with the SIFT algorithm, it retains the excellent performance of the SIFT operator but solves the problems of high complexity and time consumption of the SIFT algorithm, mainly in its removal of the down-sampling process and changing the size and scale of the original Gaussian template, thus improving the processing speed [29]. The process of image alignment is to detect the feature points of the source image and the image to be aligned and the corresponding descriptors using the SURF algorithm, and then obtain the transformation matrix of the image to be aligned relative to the source image, project all these sub-region images under the same reference coordinate system, and their corresponding pixel values form a new image and complete the SR reconstruction by the convex set projection method [7,30].

### 2.3. Field of View (FOV) Extension Method

Imaging field of view extension is a technique that combines the FOV of a single camera with an image stitching algorithm under RPIS. Sub-area position control of the image sequence for the extended FOV is possible in RPIS because the Risley prisms have the flexibility to control the precise pointing of the camera’s imaging view axis. The algorithm for solving the Risley prisms for accurate pointing of the target in the object plane has been described in detail in the literature [28]. In contrast, the literature [31] only introduces the theoretical realization of the maximum FOV without a detailed representation of the imaging location and the number of image acquisitions, so in this paper, we plan the location and number of FOV extension sub-regions reasonably for the coverage FOV and camera FOV of the RPIS to avoid the redundancy of image data in the process of realizing a larger extension of FOV to improve the efficiency of FOV extension. As shown in Figure 5, we assume that the camera imaging FOV is M × N with pixel pitch *p*. The imaging extended sub-regions can be represented by deflection angle E and azimuth angle A. $SL_{x}$, and $SL_{y}$ denotes the lateral and vertical angles in the extended FOV, respectively.
(5)$$\begin{array}{l} SL_{x}=\frac{2N^{2}}{M^{2}+N^{2}}\cdot E_{\max}\\ SL_{y}=\frac{2M^{2}}{M^{2}+N^{2}}\cdot E_{\max} \end{array}$$
(6)$$\begin{array}{l} sl_{x}=\frac{\Theta_{x}}{num_{x}}\\ sl_{y}=\frac{\Theta_{y}}{num_{y}} \end{array}$$
(7)$$\begin{array}{cc} {\mathrm{num}}_{x}= & \left\{ \begin{array}{ll} 0, & SL_{x}<\Theta_{x}\\ ceil(\frac{SL_{x}}{\Theta_{x}}), & SL_{x}\geq \Theta_{x} \end{array}\right.\\ {\mathrm{num}}_{x}= & \left\{ \begin{array}{ll} 0, & SL_{y}<\Theta_{y}\\ ceil(\frac{SL_{y}}{\Theta_{y}}), & SL_{y}\geq \Theta_{y} \end{array}\right. \end{array}$$
where $\Theta_{x}=\arctan\frac{p\cdot N}{f}$ is the horizontal angle of the camera and $\Theta_{y}=\arctan\frac{p\cdot M}{f}$ is the vertical angle of the camera, $E_{max}$ denotes the maximum deflection angle of the imaging visual axis, where, $num_{x}$, $num_{y}$ denotes the number of sub-regions along the x and y directions in the FOV extension, respectively, and the extended lateral step $sl_{x}$ and longitudinal step $sl_{y}$ are obtained from Equations (6) and (7), so as to obtain the azimuth angle A and deflection angle E at each position, and the corresponding (θ_1_,θ_2_) at each position is obtained by combining with the exact inverse solution, so as to complete the position calculation of the FOV extension.

In this paper, the FOV extension is used to extract highly different features from the image data using the scale-invariant feature transform (SIFT) method [29]. SIFT feature extraction is manifested in four main stages: scale space extremum monitoring, key point localization, orientation assignment, and key point descriptors. Random sample consistency (RANSAC) is used in the feature point matching process to exclude outliers and improve the correctness of the transformation matrix and matching speed. Finally, a multi-band image fusion algorithm is used to project the images of all locations into a unified coordinate system for stitching to obtain a larger image.

## 3. Simulation and Results

The relevant parameters of RPIS are shown in Table 1. The lens’s optical aperture (pupil) diameter D=f/F=8 mm. The Nyquist frequency of the image sensor is derived as $v_{p}=1/2p=72.5\;{\mathrm{mm}}^{-1}$ and the optical spatial cutoff frequency of diffraction limit $v_{o}=D/1.22\lambda f=621\;{\mathrm{mm}}^{-1}$, so it can be introduced that the resolution reconstruction factor $h\leq v_{0}/v_{p}=8$ is achievable.

### 3.1. Modeling Validation

The imaging view axis sub-pixel scan was simulated using a Risley prisms, assuming h = 5. The four optical axis points of the initial rotation angle were used to calculate the location points of sub-pixel displacement, and the theoretical optical axis point positions were compared with the actual optical axis point positions. As shown in Figure 6, the adjacent pixel points are divided into h × h equally spaced sub-pixel points, and the alignment accuracy of the imaging view axis is relative to the camera imaging system. In order to measure the error between the actual optical axis point position and the theoretical point position, we define the pixel difference between the actual viewpoint position and the theoretical viewpoint position as the Alignment Pixel Error (APE) and use it as a judgment of the imaging viewpoint pointing accuracy. In the example of Figure 6, the maximum APE is less than 0.01 among all the optical axis points.

### 3.2. Image Distortion and Correction Analysis

In the RPIS, the inconsistent angle of the incident prism of the beam in the object FOV and its nonlinear characteristics on beam deflection causes the target beam from the object FOV not to be deflected consistently, which eventually leads to the imaging distortion phenomenon. The image distortion will cause the pixel spacing in the pixel array to change, resulting in redundancy or loss of SR reconstruction data, and it is necessary to study the pixel spacing uniformity of the acquired images.

From the data in Figure 7 and Table 2, it can be concluded that the greater the deviation of the imaging visual axis from the central optical axis, the more severe the distortion of the image. The image is basically distortion-free when the view axis is close to the center of the optical axis. For an arbitrary imaging viewpoint, the pixel pitch variation gradually approaches the ideal variation value1 after distortion correction, which substantially reduces the fluctuation of pixel pitch variation. For the large-angle Risley prisms view-axis deflection imaging in this paper, the pixel pitch range of the distorted image exceeds ±1% at deflection angles greater than 3°, and the fluctuation is large, which is not conducive to sub-pixel displacement reconstruction. In contrast, using the inverse ray tracing for distortion correction, the pixel spacing range between the corrected image data is within ±0.5% and the fluctuation rate is controlled within 0.6%, and the corrected image is close to ideal imaging, which lays a good foundation for image reconstruction and stitching.

### 3.3. Imaging View Axis Pre-Point Position Analysis

In many papers using Risley scanning imaging, the literature [23,24] provides an in-depth study of the imaging FOV expansion in relation to system parameters (e.g., wedge angle *α* of the prism and refractive index *n*) and draws relevant conclusions. However, the number and location of image acquisition in imaging extension are not elaborated.

We use the parameters in Table 1 to simulate using the FOV extension method in this paper, and the results are shown in Figure 8. We note that the horizontal FOV in the original camera imaging FOV is expanded from 19.5° to 37.1°, and the vertical FOV is expanded from 13.3° to 27.9°. The expanded FOV is about 4 times larger than the original camera FOV.

## 4. Experiments and Results

To further evaluate the performance of the proposed SR imaging method and to verify the feasibility and robustness of the pre-located image extension. The experimental setup consists of a charge-coupled device (camera) camera, an optical lens, a Risley scanning system, and a filter, as shown in Figure 9. The relevant parameters of the camera, optical lens, and Risley scanning system are shown in Table 1. To achieve a fast response and precise pointing control, the two prisms in the Risley scanning system are rotated by two independent DC servo motors controlled by a computer via a serial port, and the minimum turning angle of the DC servo motor can reach 0.001° to achieve sub-pixel displacement deflection pointing accuracy. For broadband spectral imaging, the unitary prism introduces chromatic distortion. To ensure the HR imaging quality in the imaging extension, a filter with a center wavelength of 660 nm needs to be added in front of the camera to achieve narrowband imaging and reduce the chromatic distortion brought by the optical element.

### 4.1. Image Distortion Correction

Section 3 analyzes theoretically that the correction of imaging distortions caused by nonlinear deflection of the Risley prisms is feasible. In order to verify the inverse ray tracing correction method, multiple images are acquired using RPIS pointing at different view axes, and the inverse ray tracing method is used to correct the real distorted images. The experimental scheme adjusts the camera imaging view axis to three different positions to acquire three real images. The images contain 12 × 9 black and white square patterns. It is often used as an image calibration and correction to facilitate the demonstration of four-point positioning and distortion correction effects [32,33,34], and the size ratio is set to 4:3 in order to adapt to the CCD chip size.The camera parameters are shown in Table 1, and E_max_ = 12.16° according to Equation (1). The lens used in this experiment has a small effect on the imaging distortion, so the lens distortion is not corrected.

Figure 10 shows the acquired original image and the distortion-corrected image. When the imaging view axis is in the horizontal direction, the observed image is compressed in the horizontal direction, and the corresponding distortion-corrected image is stretched in the horizontal direction to reduce the horizontal direction distortion; when the imaging view axis is in the vertical direction, the original image is compressed in the vertical direction, and the corresponding distortion-corrected image is stretched in the vertical direction to reduce the vertical direction distortion; when the orientation of the imaging view axis is at 225°, the image is compressed in this direction, and the corrected image is stretched along this. The corrected image is basically distortion-free.

### 4.2. Super-Resolution Imaging

According to the analysis in Section 3, the resolution reconstruction factor h is theoretically achievable by a factor of 8 (h ≤ 8). However, considering the assembly, fabrication, and limited Risley prisms rotation angle control, the reconstruction factor is set to 5.

First, the U.S. Air Force (USAF1951) resolution plate was chosen to test the resolution limit of the camera, and to make the imaging quality better, a white light transmitter was added behind the resolution plate to provide uniform back light. The SR imaging results are shown in Figure 11. The middle image is the SR image of the red area, and the right image is the magnified portion of the red area of the LR image. The black and white stripe groups marked by yellow and magenta dashed lines indicate the gray gradient variation of the black and white stripe groups of the same element in the same group. In the HR image, the grayscale variation of the black-and-white stripe group in the first element of the second group in the HR image varies periodically with equal amplitude, while the grayscale amplitude variation in the LR image is larger and is the smallest stripe group that can be distinguished in the LR image. The black-and-white stripe group in the fourth element of the third group in the resolution plate is visible in the HR image with a periodic variation of grayscale change, but with a large fluctuation of amplitude, and is the smallest stripe group that can be distinguished in this image. Checking the resolution of the USAF1951 resolution plate we can get the resolution (number of line pairs) of the second group 2nd element as 4.49 lp/mm and the resolution of the third group 4th element as 11.31 lp/mm. The results show that the scene resolution of the SR image is increased by 2.5 times, which depends on the ratio of the resolution corresponding to the smallest stripe group that can be resolved in the HR image and the LR image. The reduction of the resolution reconstruction factor is mainly due to the mounting error, and the object distance measurement error. In addition, optical axis alignment, prisms relative zero calibration, and camera calibration all cause an increase in sub-pixel scanning error, which indirectly causes a decrease in the quality of SR reconstruction. However, compared to reference [7], which used a 5 × 5 camera array and achieved a resolution enhancement factor of 2.25, the proposed method uses a single camera to achieve a resolution reconstruction factor of 2.5, showing the potential for low-cost and efficient SR applications.

Figure 11 SR imaging results of the (USAF1951) resolution plate. The left image is the SR reconstruction image of the resolution plate. The left image is its HR image corresponding to the red region, and the right image is its LR image of the resolution plate corresponding to the red region.

Next, we conducted a SR imaging experiment on the indoor scene, and the results are shown in Figure 12. From the figure, we can see that more details of the scene are recovered after the SR reconstruction. In the green border, the blended black and white stripes in the LR image are recovered in the SR image. The Chinese font in the blue and red areas of the SR image becomes recognizable, while it is blurred and unrecognizable in the corresponding LR image. In addition, the higher pixel resolution reduces the artifacts caused by pixels.

Third, to further test the quality of the SR reconstructed images of the proposed method, we calculated the modulation transfer function (MTF) of the prototype using tilted-edge imaging in the literature [35]. For comparison, we used LR images by bilinear interpolation to obtain images with the same resolution as the SR (POCS) images. Figure 13 shows the MTF of the SR image versus the bilinear interpolated image. The results show that the SR image using the convex set projection method has higher quality compared to the bilinear interpolated image.

### 4.3. FOV Extension

The deflection angle E and azimuth angle A of each sub-region is obtained according to the FOV expansion method elaborated in Section 2, and θ_1_ and θ_2_ are calculated using the exact inverse solution, as shown in Table 3. The camera imaging view axis is scanned in a clockwise spiral method as described in the previous section to improve the image sampling efficiency. Once the two prisms are rotated to the desired E and A, the camera acquires all sub-region images at the corresponding viewpoints, as shown in Figure 14, where the maximum deflection angle of the imaging view axis of the camera is up to 11.62° in this experiment.

We perform SR reconstruction and FOV extension experiments of outdoor scenes to validate the proposed imaging method, where Figure 14a shows the stitched images of the prototype, including the stitching positions of the 9 sub-images (colored dashed boxes) and the final cropped images (yellow solid boxes). In the 9 images to be stitched, we use SIFT feature extraction, combined with the RANSAC algorithm to select the correct feature matching. The transformation matrix between each pair of images is obtained, and the images of all sub-regions are converted to a uniform coordinate system for mutual alignment by matrix multiplication. Finally, the overlapping area is smoothed based on the linear mixing strategy, and a composite image of 6592 pixels × 4540 pixels is obtained by cropping along the yellow solid frame. The FOV of the whole stitched image is 43.5° × 30.6°, the FOV of the cropped image is 35.7° × 22.4° and the FOV of the original camera is 19.5° × 13.3°. The size of the covered area is three times the size of the original camera’s FOV, and the FOV expansion ratio in the experimental results is slightly smaller than the theoretical value, and the deviation may be caused by the system installation error and the stitching algorithm error. In Figure 14b, the SR image is shown in the center of the extended FOV, and the two pairs of local areas are located in the upper left and lower right areas of the SR image. The flower pot and the fence in the lower right of the middle are recovered in the SR image.

## 5. Conclusions

In summary, in order to realize SR imaging, a beam inverse tracking method is proposed to correct the image combined with sub-pixel SR reconstruction, which solves the problem of large pixel spacing variation and high fluctuation rate due to large angular deflection of the Risley prisms and lays a good foundation for subsequent SR reconstruction and FOV expansion. According to the inference, the reconstruction factor of the system is basically only related to the degree of blending of the imaging system. Meanwhile, in the FOV extension, a method based on the image acquisition position, order, and the number are proposed to improve the image acquisition efficiency and reduce the image stitching data redundancy. In the future, the method will be combined with a more adaptive imaging strategy that uses a Risley prisms scanner to generate camera-specific view-axis motion trajectories. The experimental results of SR reconstruction and FOV expansion show that the reconstructed scene resolution can be increased by a factor of 2.5 and the FOV can be expanded by a factor of 3. The SR reconstructed image quality has a significant advantage over the reconstructed images using bilinear interpolation. In terms of planning camera view axis positions and paths, the method will probably be combined with a more adaptive imaging strategy to generate specific camera view axis motion trajectories, improving a promising solution for various subsequent scanning imaging applications. For SR reconstruction, sub-pixel acquisition using the Risley prisms shows greater potential in terms of low cost, high efficiency, and central concave imaging that is unusually long or complex.

## Figures and Tables

**Figure 1 sensors-23-01793-f001:**
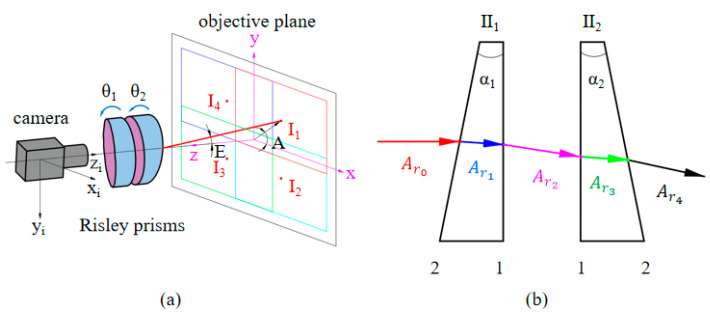
RPIS imaging system: (**a**) is the RPIS imaging system and (**b**) is the structure 21-12 Risley prisms planar model.

**Figure 2 sensors-23-01793-f002:**
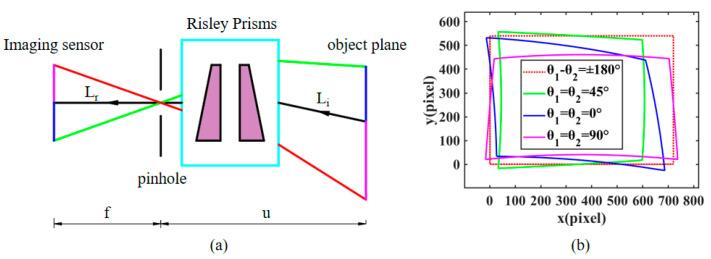
Effect of Risley prisms on imaging: (**a**) Beam deflection model under RPIS. (**b**) The effect of different rotational combinations on the imaging FOV.

**Figure 3 sensors-23-01793-f003:**
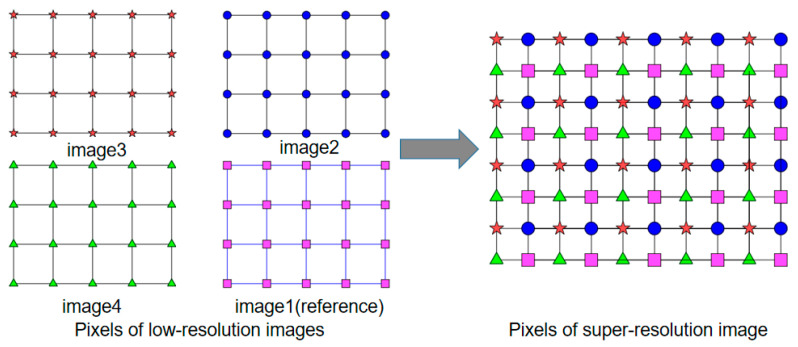
Principle of SR reconstruction of LR sub-pixel shift sequence images.

**Figure 4 sensors-23-01793-f004:**
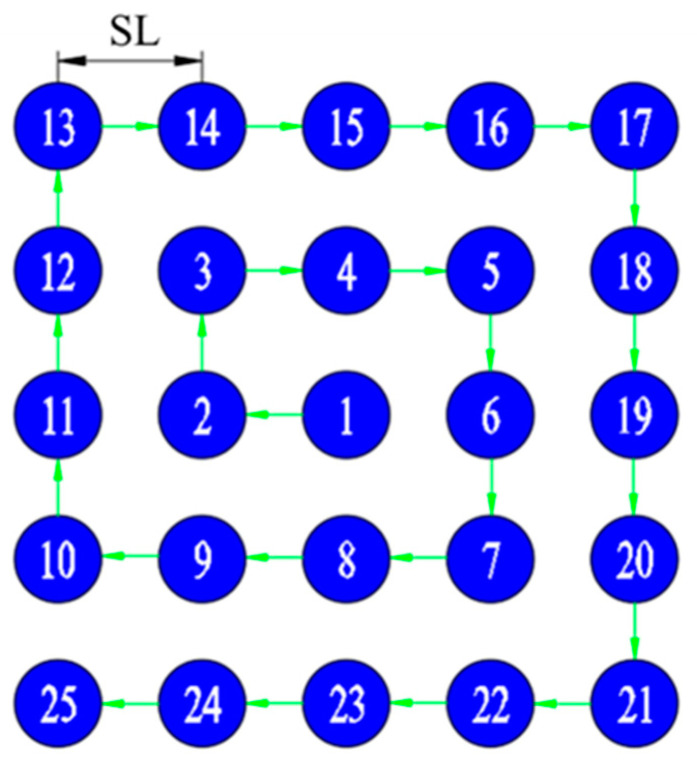
Sub-pixel scanning imaging path (clockwise spiral scanning method).

**Figure 5 sensors-23-01793-f005:**
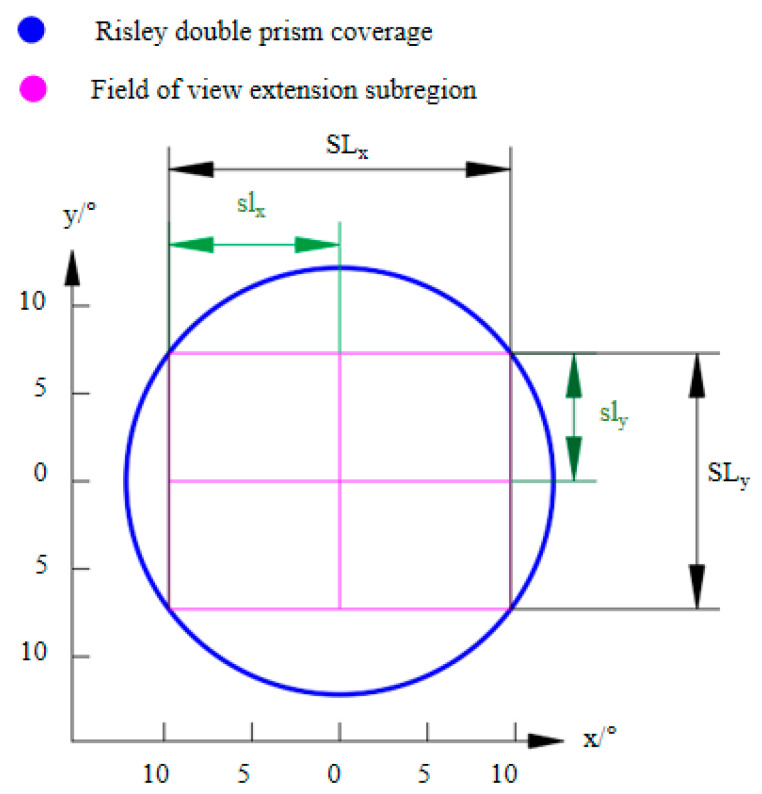
Imaging extension sub-region location planning.

**Figure 6 sensors-23-01793-f006:**
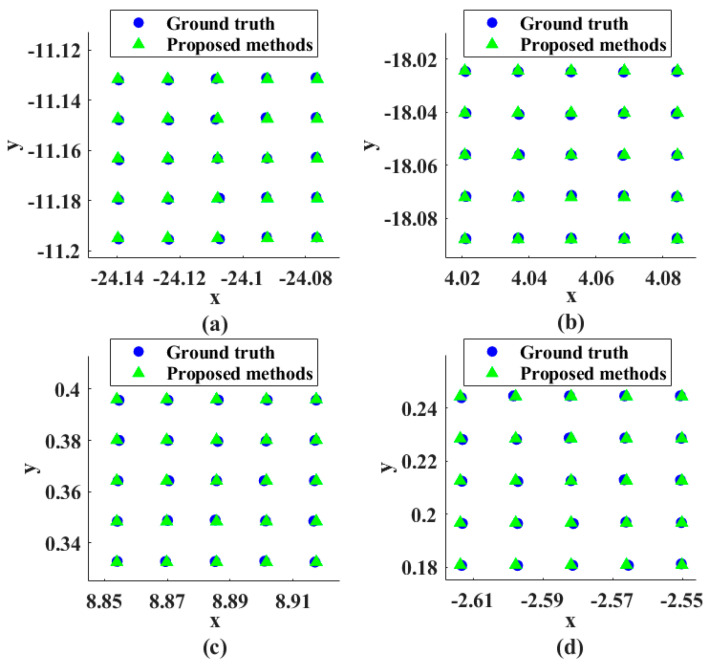
Comparison of the actual sub−pixel points with the theoretical optical axis points for the imaging view axis simulation. The initial rotation angles (θ_1_,θ_2_) of (**a**–**d**) are (20°, 30°), (60°, 150°), (120°, 260°) and (50°, 240°), respectively.

**Figure 7 sensors-23-01793-f007:**
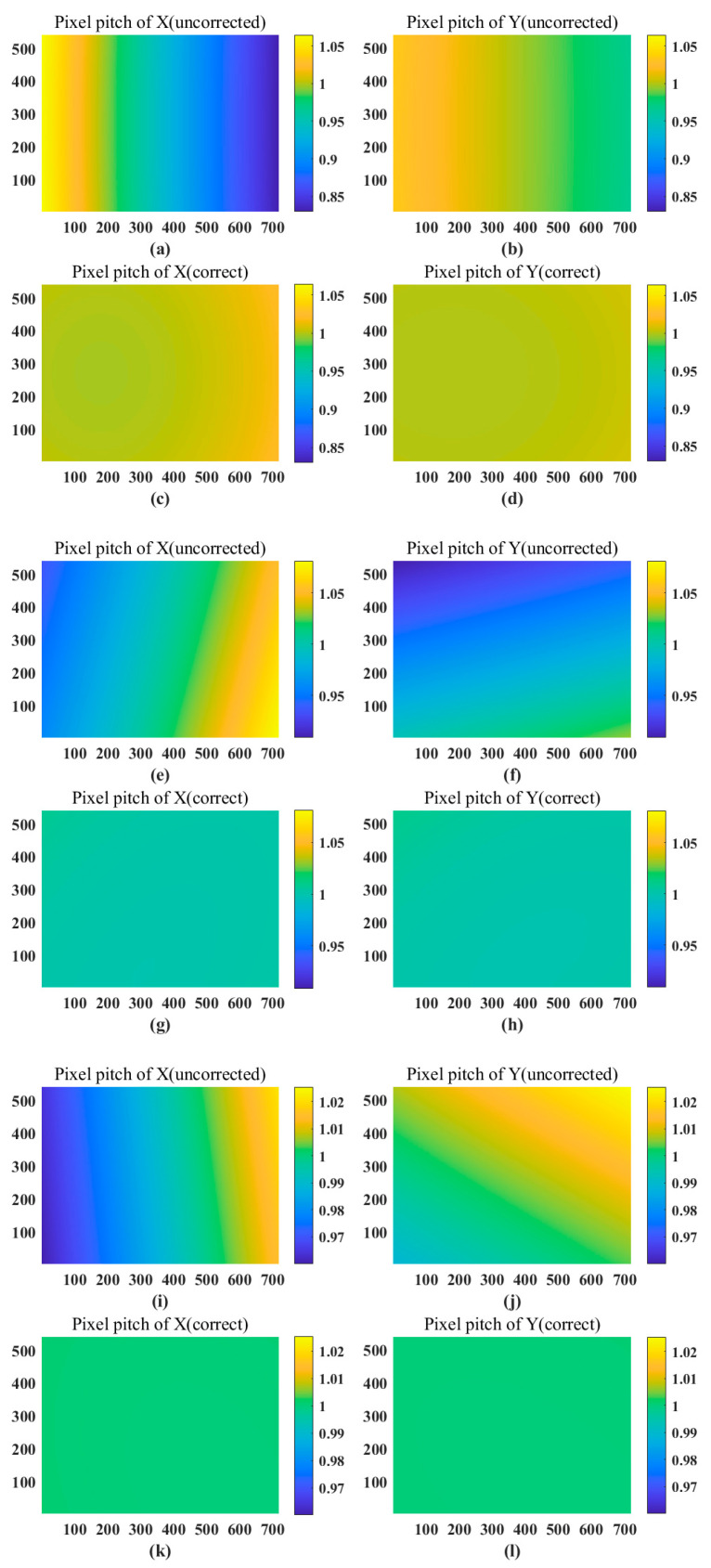
The distorted images at different positions correspond to the change in pixel spacing of the corrected images. In this example, f = 16 mm, n = 1.51421, α = 11°21′, p = 6.9 um, M = 540, N = 720 (**a**–**d**) (θ_1_, θ_2_) = (0°, 0°) distorted image corresponds to the pixel pitch variation of the corrected image in x and y directions. (**e**–**h**) (θ_1_, θ_2_) = (90°, 180°) the distorted image corresponds to the pixel pitch change in the x and y directions of the corrected image. (**i**–**l**) (θ_1_, θ_2_) = (130°, 280°) the distorted image corresponds to the pixel pitch change in the x and y directions of the corrected image.

**Figure 8 sensors-23-01793-f008:**
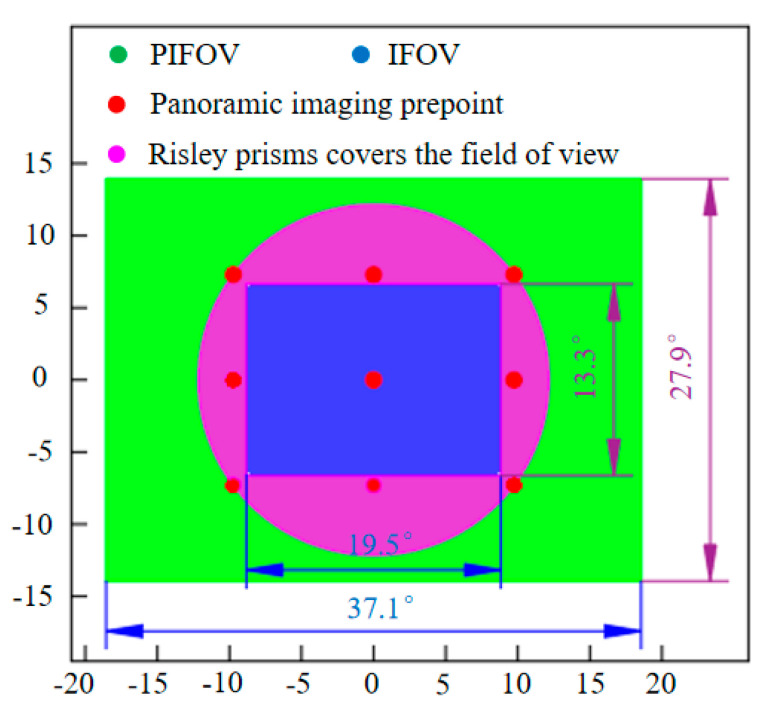
FOV, imaging field of view (IFOV), imaging pre−point position, and Risley prisms coverage FOV.

**Figure 9 sensors-23-01793-f009:**
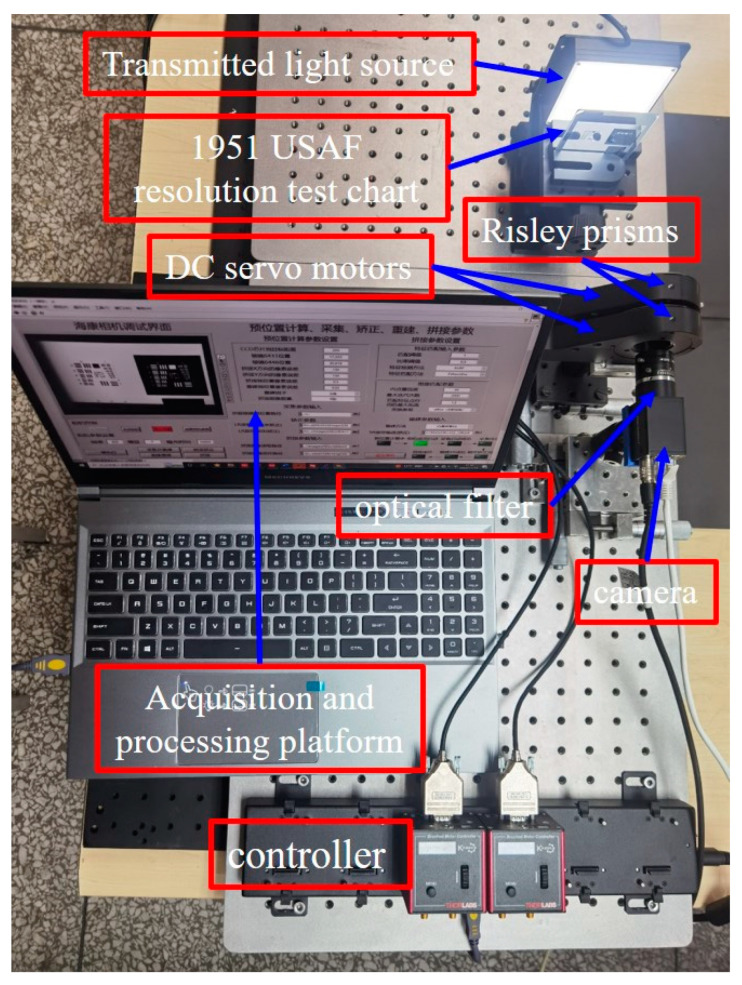
Experimental setup.

**Figure 10 sensors-23-01793-f010:**
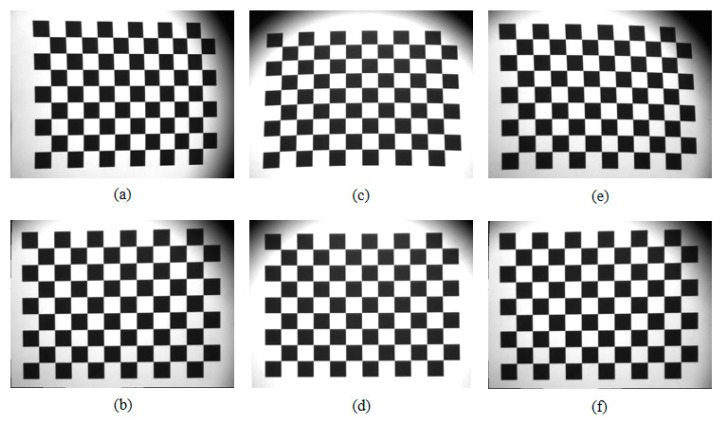
Imaging distortions caused by the Risley prisms and their correction. (**a**) is the image taken with the imaging view axis at A = 180° and E = 12.16°, (**b**) is its corresponding distortion-corrected image, (**c**) is the image taken with the imaging view axis at A = 270° and E = 12.16°, (**d**) is its corresponding distortion-corrected image, (**e**) is the image taken with the imaging view axis at A = 225° and E = 12.16°, (**f**) is its corresponding distortion-corrected image.

**Figure 11 sensors-23-01793-f011:**
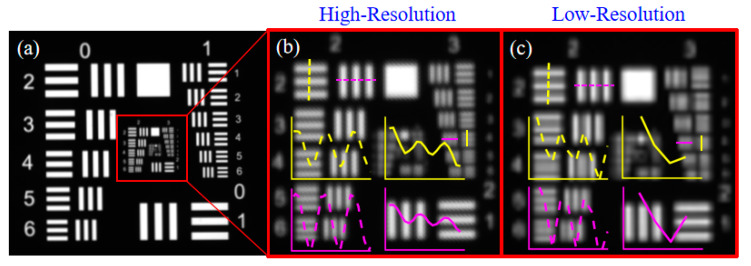
(USAF1951) Comparison of SR imaging of the resolution plate with the source camera imaging results. (**a**) SR reconstructed image using SR sub-pixel scanning technique; (**b**) enlarged middle view of SR reconstructed image; (**c**) enlarged original camera image of the same area as (**b**).

**Figure 12 sensors-23-01793-f012:**
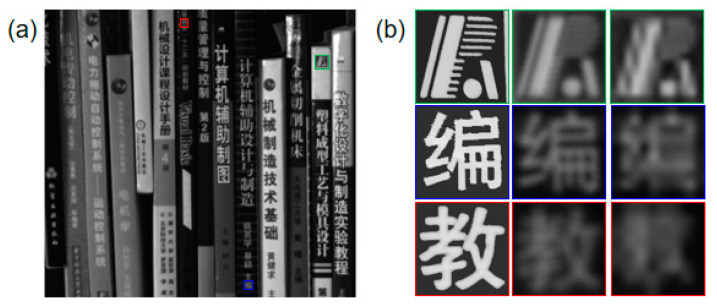
SR imaging experiment. (**a**) SR reconstructed image is shown. (**b**) Local magnification regions correspond to red, green, and blue in the SR reconstructed images. For each set of local magnification images, the left column of images is a close-up of the three local areas corresponding to red, green, and blue in (**a**) image taken by the camera at high magnification, the middle column of images is from the SR reconstructed image, and the right column is from the LR image.

**Figure 13 sensors-23-01793-f013:**
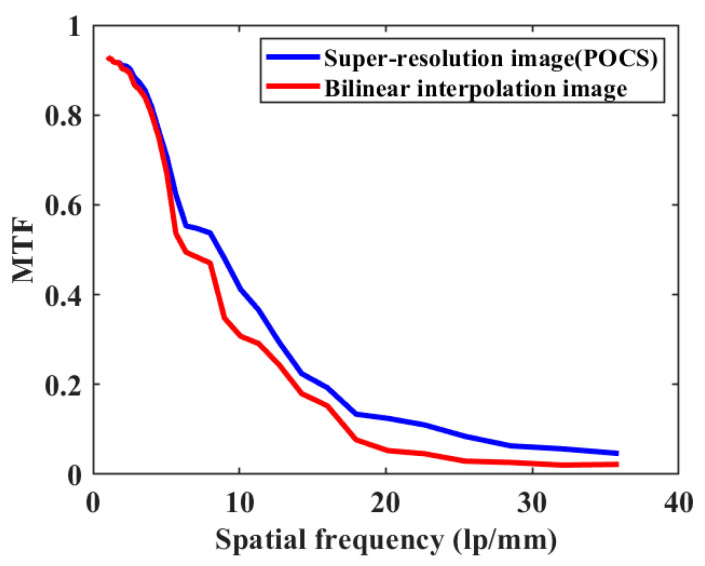
MTF of bilinear interpolated image and SR reconstructed image by convex set projection method.

**Figure 14 sensors-23-01793-f014:**
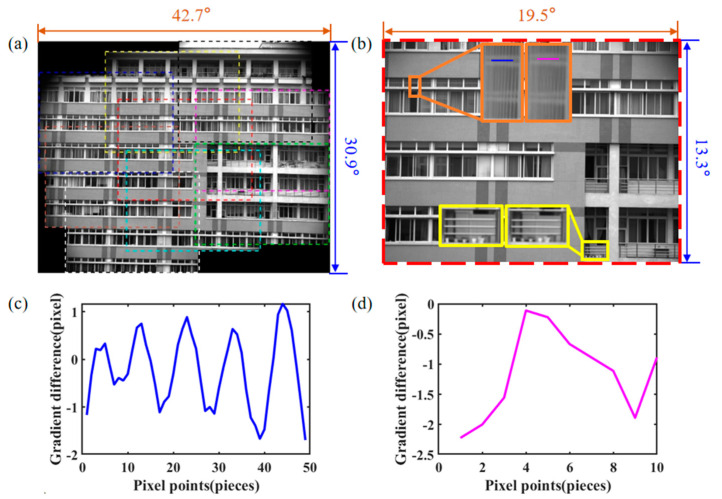
Stitched image and the central SR image with extended FOV, (**a**) colored dashed boxes with dotted lines in (**a**) indicate each of the nine stitched images, (**b**) colored solid boxes indicate the local area of the central SR image, with the SR image segment on the left and the LR image on the right, (**c**,**d**) are the intensity gradient changes at the corresponding blue and magenta positions in the orange boxes in (**b**), respectively.

**Table 1 sensors-23-01793-t001:** Main parameters of the simulated RPIS.

Parameter	Abbreviation	Value
Pixel	*p*	6.9 um
Rows × Cols of pixel array	M × N	540 × 720
Focal length	F	16 mm
F-number	*f*	2
Wedge angle	*α*	11°21′
Object distance	*u*	220 mm
Refractive index	*n*	1.51421

**Table 2 sensors-23-01793-t002:** Mean and variance of pixel pitch variation for simulated images at different locations.

Position	Distort	Correct
(0°, 0°)	u_x_: 0.9424, σ_x_: 0.0669 u_y_: 1.0001, σ_y_: 0.0192	u_x_: 1.0041, σ_x_: 0.0059 u_y_: 1.0014, σ_y_: 0.0017
(90°, 180°)	u_x_: 1.0044, σ_x_: 0.0353 u_y_: 0.9681, σ_y_: 0.0272	u_x_: 1.0014, σ_x_: 0.0014 u_y_: 1.0014, σ_y_: 0.0020
(130°, 280°)	u_x_: 0.9903, σ_x_: 0.0161 u_y_: 1.0060, σ_y_: 0.0074	u_x_: 1.0002, σ_x_: 0.0002 u_y_: 1.0001, σ_y_: 0.0001

**Table 3 sensors-23-01793-t003:** Deflection position of imaging view axis and prism rotation angle of sub-region images.

	A/°	E/°	θ_1_/°	θ_2_/°
(a)	89.980	0.205	359	181
(b)	178.735	9.254	36.678	319.974
(c)	142.378	11.621	338.608	306.791
(d)	90.001	7.156	323.564	217.589
(e)	37.625	11.620	233.884	202.010
(f)	1.256	9.252	221.215	142.479
(g)	323.991	11.383	124.171	163.046
(h)	269.999	6.701	34.091	144.789
(i)	216.007	11.385	16.212	55.036

## Data Availability

Not applicable.

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
