# Peer review of "Super-Resolution and Wide-Field-of-View Imaging Based on Large-Angle Deflection with Risley Prisms"

_sensors, 2023, doi:10.3390/s23041793_

Round 1

Reviewer 1 Report

1.     In the section 4.1 Image distortion correction. The author use images which contain 12 × 9 black and white square patterns. If the image has other patters, do the results support the conclusions?

2.     The author should compare the result with other work to explain the advantages of the proposed technique.

3.     The resolution of the figures is low, such as Fig. 13. The font sizes are not uniform, such as Figs. 1, 2, 5 and 8. The author should check the style carefully.

4.     The description of Fig. 14(c) is missing. What are the meaning of the XY axes? And the legend and scale?

Reviewer 2 Report

Dear Dr. Leng,

I have finished reviewing the article entitled “Super-resolution and wide-field-of-view imaging based on large-angle deflection with Risley prisms” by C. Gui et al.

In this document, the authors describe how adding Risley prisms to a single camera allows super-resolution and extends the field of view after making image processing.

I have some comments on this article.

When we usually speak about super-resolution, we must prevent the readers from the kind of super-resolution we are talking about. Indeed, in an optical system, we find the optical super-resolution related to the optical elements, I mean, the lenses. In such a case, the diffraction limit prevents going beyond the angular resolution described by θ=1.22λ/D, where λ is the wavelength and D is the lens diameter. Breaking this limit in microscopy was recompensed by the 2014 Nobel prize in Chemistry. Three main techniques were used to break the limit: (a) the principle for stimulated emission depletion STED microscopy (V. Westphal, et al., “Video-rate far-field optical nanoscopy dissects synaptic vesicle movement,” Science320(5873), 246-249 (2008)), (b) the photoactivated localization microscopy (E. Betzig and J. R. Chichester. “Single molecules observed by near-field scanning optical microscopy,” Science 262(5138), 1422-1425 (1993)), and (c) single-molecule spectroscopy (W. E. Moerner & L. Kador, “Optical detection and spectroscopy of single molecules in a solid,” Physical review letters62(21), 2535 (1989)). A different experimental approach is demonstrated in: A. Aguilar et al., “Super-resolution with a complex-amplitude pupil mask encoded in the first diffraction order of a phase grating,” Opt. Lasers Eng., 134, 106247 (2020); applies to any optical system and is not restricted only to microscopy.

The second use of the super-resolution term is in sensors, mainly in solid states devices such as CCDs or CMOS used in optoelectronic cameras. In this case, the sensor resolution is expressed, for instance, in line pairs per millimeter, lines, and MTF. The smaller the pixel results in a broader MTF curve and, thus, in better detection of higher frequency energy. In sensors, the Geometrical Super Resolution means that the resolution of digital imaging sensors is enhanced. So there is a difference between super-resolution in the optical system and super-resolution in a sensor. The resolution limit is effectively broken to obtain the super-resolution in the first case. In contrast, in the second case, the resolution is enhanced. The authors must make this difference in the introduction to avoid misleading use of super-resolution for the enhanced resolution of a sensor.

Concluding: I ask the authors to describe the context of sensors’ geometrical super-resolution (or sensors’ resolution enhancement) in the introductory section. A brief description between optical super-resolution and sensors’ super-resolution will allow readers to avoid confusion.

Reviewer 3 Report

This manuscript, a beam inverse tracking method is proposed to correct the image combined with sub-pixel SR reconstruction. The manuscript has a proper structure. The results are described in detail and are sufficiently innovative to be published. Therefore, I recommend the publication of this manuscript in the Sensor Journal.
